

# A mini-review of the relationship between intestinal microecology and acute respiratory distress syndrome

Lujia Chen[1], Hao He[1], Cheng Li[1], Sha Nie[1], Dong Liu[1] and Qiwei Shi[2]

[1] Department of Respiratory and Critical Care Medicine, The Fourth Hospital of Changsha, Changsha, Hunan, China
[2] Microbiological Testing Laboratory, Hengyang Center for Disease Control and Prevention, Hengyang, Hunan, China

## ABSTRACT

Acute respiratory distress syndrome (ARDS), a critical condition with high mortality, arises from dysregulated inflammation and lung injury. While evidence-based supportive care remains foundational, the lack of effective targeted therapies underscores the need for novel approaches. This review focuses on the emerging role of intestinal microecology in ARDS pathogenesis *via* the gut-lung axis. We discuss how ARDS disrupts gut barrier integrity, promotes dysbiosis and bacterial translocation, and highlight the significance of some gut microbiota-derived metabolites in modulating pulmonary immunity and inflammation. Furthermore, we explore how intestinal microecology influences ARDS progression through mechanisms like oxidative stress, apoptosis, autophagy, and pyroptosis. The review also examines the potential of microecology-based interventions and draws insights from failed immunomodulatory trials, emphasizing the critical interplay between the microbiome and host immunity. By synthesizing these links, this review identifies the gut microbiota as a source of potential early-warning biomarkers and novel therapeutic targets, aiming to inform future strategies for managing ARDS in the intensive care unit (ICU).

## INTRODUCTION

Acute respiratory distress syndrome (ARDS) is a clinical syndrome characterized by intractable hypoxemia caused by intra-pulmonary (*e.g.*, pneumonia, aspiration) and/or extra-pulmonary (*e.g.*, sepsis, trauma) factors. Its pathophysiology is characterized by acute, diffuse, and acute lung injury (ALI), which leads to increased permeability of alveolar capillaries, pulmonary edema, alveolar collapse, and damage to lung tissue (*Kain, Dionne & Marshall, 2024*). Current management of ARDS prioritizes evidence-based supportive therapies, particularly lung-protective ventilation strategies and adjunctive interventions such as prone positioning for moderate-to-severe cases. However, there remains a critical lack of "targeted pharmacological therapies" proven to effectively modulate the dysregulated inflammatory response underlying ARDS pathogenesis (*Qadir et al., 2024*). Although the optimization of mechanical ventilation therapy in recent years

Corresponding author
Qiwei Shi, qiweis11@outlook.com

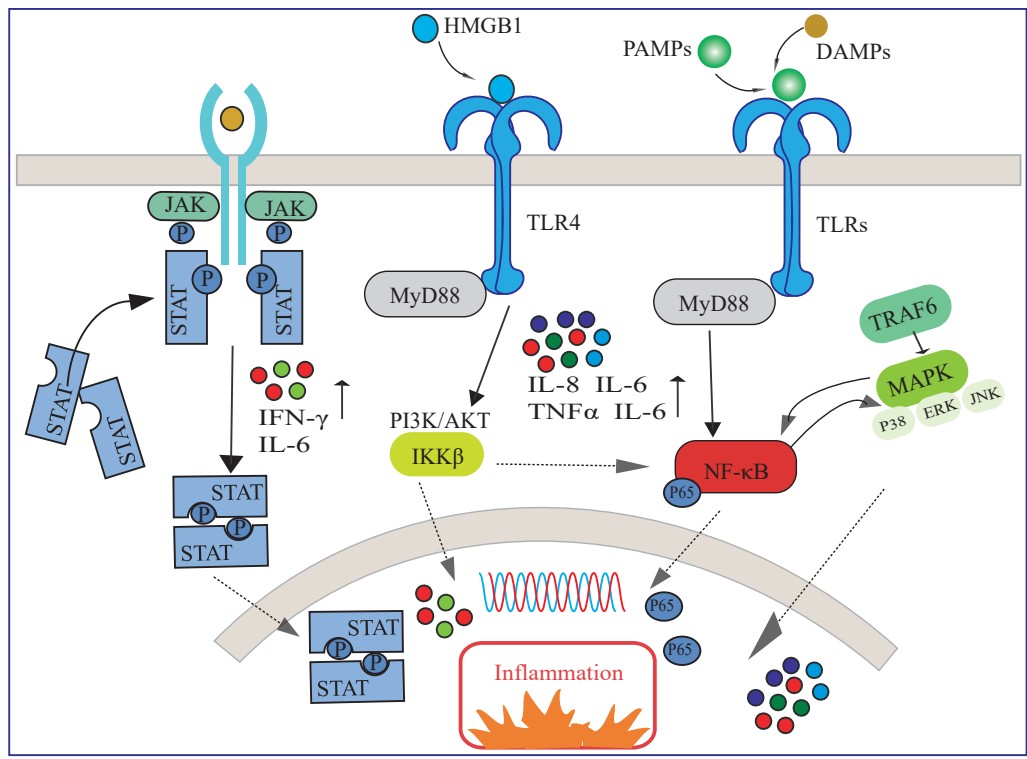

**Figure 1** **Several inflammatory pathways involved in ARDS.** HMGB1, high-mobility group box 1 protein; DAMPs, damage-associated molecular pattern; TLR, toll-like receptor; PAMPs, pathogen-associated molecular patterns; NF-κ B, nuclear factor-κ B; MAPK, mitogen-activated protein kinase; IL, interleukin; TNF-α, tumor necrosis factor-α; IFN-γ, interferon-γ.

has improved the prognosis of ARDS patients to a certain extent, the in-hospital mortality rate of approximately a million ARDS patients per year worldwide still exceeds 30% (*Gorman, O'Kane & McAuley, 2022*). According to survey data from over 50 countries, 10.4% (95% confidence interval CI [10.0–10.7]%) of intensive care unit (ICU) patients suffered from ARDS. When patients progress to severe ARDS, the mortality rate rises above 40.0% (*Bellani et al., 2016*). Due to prolonged hospital stays, a difficult-to-treat condition, and high treatment costs, ARDS significantly increases the socioeconomic burden. Therefore, exploring the pathogenesis of inflammatory injury in ARDS will help develop fundamental treatments, improve therapeutic efficacy, shorten the treatment period, and save social costs.

In response to lung injury, the immune system initiates an inflammatory response to clear pathogens and protect lung tissue, and a balance of immune regulatory mechanisms is essential for mitigating lung injury and promoting lung function recovery. Available evidence suggests that several inflammatory pathways are involved in activating immune cells and releasing inflammatory mediators (Fig. 1). Understanding the mechanisms of action of these inflammatory pathways is essential for elucidating their contributions to ARDS recovery and identifying potential therapeutic targets (*Matthay & Zemans, 2011*).

As shown in Fig. 1, upon pathogen invasion, multiple signaling pathways transmit inflammatory signals from extra-cellular to intra-cellular. The interactions among these signaling pathways constitute a complex network, collectively regulating the immune and inflammatory responses in ARDS (*Zheng et al., 2024*). HMGB1 is a damage-associated molecular pattern (DAMP) that activates inflammatory responses by binding to toll-like receptor (TLR) 4 and other receptors (*Li et al., 2020*). TLRs are crucial receptors for recognizing pathogen-associated molecular patterns (PAMPs) and DAMPs (*Khanmohammadi & Rezaei, 2021*). In ARDS, TLRs can transmit downstream signals through MyD88-dependent or independent pathways, leading to the nuclear translocation of NF-κB, or activate members of the mitogen-activated protein kinase (MAPK) family such as p38, ERK, and JNK through TRAF6-mediated pathways, thereby increasing the release of inflammatory cytokines such as IL-1, IL-6, IL-8, and TNF-α. These cytokines have been proven to play a key role in the pathogenesis of ALI/ARDS (*Tolle & Standiford, 2013*; *Wu et al., 2018*). Specifically, there may exist a positive feedback regulation between the NF-κB signaling pathway and the MAPK signaling pathway (*Feng, Sun & Li, 2015*). Additionally, the PI3K/Akt pathway is also involved in regulating the activation of NF-κB through phosphorylation of IKKβ, collectively exacerbating the inflammatory response and damage in lung tissue (*Finnberg & El-Deiry, 2004*). Furthermore, various cytokines such as IFN-γ and IL-6 participate in regulating the inflammatory response by activating the JAK/STAT signaling pathway (*Luyt et al., 2020*).

Numerous previous studies have explored the inflammatory mechanisms underlying the development and progression of ARDS induced by various precipitating factors such as systemic infections, pneumonia, aspiration, trauma, etc. Unfortunately, neither the research focused on the host (*e.g.*, genetic polymorphisms, inflammatory phenotypes) nor the studies on exposure factors (*e.g.*, virulence of pathogens, host susceptibility) have achieved breakthroughs in this field (*Meyer, Gattinoni & Calfee, 2021*). Identifying new entry points and uncovering the hidden causes of ARDS will provide a strong basis for precise and targeted therapies.

## SURVEY METHODOLOGY

We conducted a literature search on PubMed (MeSH), Google Scholar, and CNKI databases for research progress on the gut microbiota in patients with ARDS. The search keywords included "intestinal microecology", "acute respiratory distress syndrome", "ARDS", "acute lung injury", "gut microecology" in combination with, "gut microbiota", "gut microbes", "gut microbiome", "inflammation", "dysbiosis", "immune system", "inflammatory pathway", "intestinal mucosal barrier", "intestinal permeability", "gluco-corticoid", "corticosteroid", "methylprednisolone", "hydrocortisone", "ketoconazole", "convalescent plasma", "immune plasma", "immunomodulat", "immunosuppress", "anti-inflammatory", "failure" and "no benefit" which were then cross-referenced using Boolean operators "OR" and "AND" for further retrieval. Studies were screened according to the following pre-defined criteria:

Inclusion criteria: (i) Population: Human subjects diagnosed with ARDS (based on Berlin Definition or clinical diagnosis). (ii) Exposure/Intervention: Studies investigating

gut microbiota composition, dysbiosis, or related mechanisms (*e.g.*, inflammation, immune response, barrier function). (iii) Outcomes: Reported data on gut microbiota changes, inflammatory biomarkers, or clinical outcomes linked to microbiota. (iv) Study design: Clinical trials, cohort studies, case-control studies, or mechanistic human studies.

Exclusion criteria: (i) Reviews, editorials, conference abstracts without original data. (ii) Studies not focused on ARDS or microbiota. (iii) Non-English/Non-Chinese publications. (iv) Duplicate publications or data from the same cohort.

Titles/abstracts were independently screened by two authors using the above criteria. Full texts of potentially eligible articles were assessed, with disagreements resolved by a third reviewer. After screening, 103 high-quality articles met all inclusion criteria and were included.

## Intestinal microecology

Intestinal microecology has been an important area of research in recent years due to its crucial role in maintaining the normal function of the body (*Fu et al., 2022*). In normal physiological conditions, there is a stable symbiosis between the human body and the gut microbiota. However, during disease states, notable changes occur in the quantity and types of gut microbiota, especially drastic alterations observed in critically ill patients (*Dickson, 2016*). The blows of shock, respiratory failure, or multi-organ dysfunction in critically ill patients may lead to intestinal ischemia, hypoxia, or reperfusion injury, which can damage the integrity of the intestinal mucosal barrier, thus causing intestinal microecological disorders. *Petrilla et al. (2024)* compared fecal samples from critically ill patients and healthy populations, and the results showed that the abundance of Firmicutes and Bacteroidetes was lower in the gut microbiota of the experimental group, while the abundance of Proteobacteria increased. Significant changes in the patient's gut microbiota occur within 6 h of the onset of critical illness, characterized by a notable decrease in beneficial bacteria such as obligate anaerobes and *Lactobacillus*, and an increase in harmful bacteria such as *Enterococcus* and *Pseudomonas* (*Kain, Dionne & Marshall, 2024*).

Besides alterations in microbial diversity, changes in phenotype and virulence may also occur within the gut microbiota. Studies have shown that sepsis can promote the intestinal colonization of *Klebsiella pneumoniae* strains carrying the SHV-18 resistant gene that produce extended-spectrum β-lactamases, and facilitate the transfer of resistant genes to potential endogenous pathogens during antibiotic treatment (*Guan et al., 2014*). Furthermore, the gut microbiota can also translocate to other parts of the body, participating in the development of various diseases (*Manfredo et al., 2018*).

Mucin, primarily composed of the MUC2 glycoprotein secreted by goblet cells, constitutes the first line of defense against microbial invasion and serves as a major carbon source for the host (*Paone & Cani, 2020*). Notably, certain bacteria, such as *Akkermansia muciniphila* (*AKK*), not only degrade mucin but also disrupt claudin proteins, thereby compromising intestinal barrier integrity. This synergistic degradation mechanism warrants emphasis (*Bakshani et al., 2025*). Furthermore, *AKK* abundance exhibits a negative correlation with depressive phenotypes in both ARDS patients and murine ARDS models (*Zhu et al., 2025*). Additionally, under hypoxic ARDS conditions, reduced

HIF-1α expression creates a "dual assault": diminished host mucin synthesis coupled with enhanced bacterial degradation (*Ma, Yeom & Lim, 2022*). Therefore, preserving and restoring intestinal mucin barrier function and modulating associated microbiota represent promising therapeutic strategies for ARDS, potentially mitigating gut barrier injury and improving patient outcomes.

Recent research has increasingly focused on gut microbiota-derived metabolites—such as short-chain fatty acids (SCFAs), indoles and their derivatives, bile acids, and vitamins—due to their critical roles in modulating the intestinal micro-environment (*Krautkramer, Fan & Bäckhed, 2021*). Among these metabolites, SCFAs are the most abundant and beneficial, primarily comprising acetate, propionate, and butyrate (*Hays, Pfaffinger & Ryznar, 2024*). Butyrate serves as the primary energy source for colonic epithelial cells, stimulating mucin secretion and tight junction (TJ) protein expression to maintain gut barrier integrity (*Cheng et al., 2025*). Furthermore, SCFAs (notably butyrate and propionate) enter the systemic circulation, reach distal organs such as the lungs, and participate in pulmonary immune responses (*Hays, Pfaffinger & Ryznar, 2024*). Consequently, SCFAs represent key mediators within the gut-lung axis.

Antimicrobial peptides (AMPs), evolutionarily conserved molecules secreted by Paneth cells and epithelial lineages, critically shape intestinal microbiota composition. Among the most extensively studied intestinal AMPs are three primary classes: α-defensins (HD5/6), REG3γ, and cathelicidin (LL-37) (*Zhang, 2025*). REG3γ specifically binds peptidoglycan to neutralize Gram-positive pathogens, whereas LL-37 exhibits broad-spectrum antimicrobial activity against diverse bacteria and viruses (*Saini et al., 2022*; *Shin, Bozadjieva-Kramer & Seeley, 2023*). Conversely, α-defensins primarily target Gram-negative bacteria such as *Enterobacteriaceae* (*Wierzbicka-Rucińska et al., 2025*). Notably, intricate bidirectional crosstalk exists between the microbiota and AMPs. For instance, commensal microbes induce HD5 secretion through MyD88-dependent signaling, while *Parabacteroides goldsteinii*'s outer membrane protein A upregulates REG3γ expression in murine models (*Filipe Rosa et al., 2023*; *Wang et al., 2024*). Furthermore, SCFAs like propionate enhance REG3γ expression in cecal tissue and intestinal organoids (*Darnaud et al., 2018*). Dysregulation of AMPs significantly contributes to disease pathogenesis, as evidenced in Crohn's disease, where nucleotide-binding oligomerization domain 2 confers protection partly by directly modulating α-defensin expression—a process potentially involving NF-κB/MAPK pathway regulation, lysozyme sorting, and ATG16L1 recruitment (*Yang & Shen, 2021*). In ARDS, however, TLR4/NF-κB-driven overproduction of pro-inflammatory cytokines (TNF-α/IL-1β) suppresses HD5/6 expression, consequently permitting pathogenic overgrowth of organisms like *Enterococcus* and *Klebsiella spp* (*Aarbiou, Rabe & Hiemstra, 2002*). Therefore, therapeutic strategies designed to augment AMP activity—such as butyrate supplementation or histone deacetylase (HDAC) inhibition—represent promising approaches to restore microbial equilibrium in ARDS by counteracting this critical immune defect.

**Table 1 Summaries of major gut microbiota involved in ARDS.**

| Reference | Sample number | Species | Gut microbiota | Changes in ARDS |
|---|---|---|---|---|
| Hu et al. (2023) | 26AP-ARDS/39AP-nonARDS/20normal | Human | *Proteobacteria* phylum, *Enterobacteriaceae* family, *Escherichia-Shigella* genus, *Klebsiella pneumoniae* | Upregulated |
| | | | *Bifidobacterum* genus | Downregulated |
| Kong et al. (2019) | 7ARDS/4normal | Human | *Firmicutes, Acinetobacter* | Upregulated |
| | | | *Bacteroidetes, Proteobacteria* | Downregulated |
| Zuo et al. (2020) | 15ARDS-COVID-19/15normal | Human | *Coprobacillus, Clostridium ramosum, Clostridium hathewayi* | Upregulated |
| | | | *Faecalibacterium prausnitzii* | Downregulated |
| Cheng et al. (2022) | 120ARDS/120normal | Human | *Bifidobacterum, Enterococcu, Lactobacillus, Eubacterium* | Downregulated |
| Zheng et al. (2023) | 21ARDS-CAP/21normal | Rats | *Eubacterium, Barnesiella, Escherichia-Shigella Lactobacillus* | Upregulated |
| | | | *Muribaculum, Ruminococcacee_NK4A214_group, Blautia* | Downregulated |
| Li et al. (2014) | 16ALI-ARDS | Rats | *Fusobacteria, Helicobacter, Roseburia* | Downregulated |

## Impact of ARDS on the intestinal microecology

The hallmark inflammatory response in ARDS is characterized by elevated IL-6, TNF-α, and HMGB1, which actively compromises intestinal tight junction (TJ) integrity. These pro-inflammatory cytokines down-regulate the expression of occludin and zonula occludens-1 (ZO-1), thereby increasing intestinal permeability. This disruption facilitates bacterial translocation from the intestinal lumen into the systemic circulation (Zhang et al., 2020). In patients with ARDS, intestinal dysbiosis is manifested by a decrease in beneficial commensal bacteria (*e.g.*, *Faecalibacterium prausnitzii*, *Bacteroides*, and *Bifidobacterium*) and an increase in opportunistic pathogens (including certain *Enterobacteriaceae*, *Clostridium difficile*, and *Streptococcus spp.*) which primarily play crucial roles in affecting the intestinal mucosal barrier, systemic inflammatory status, and immune responses (Li et al., 2014; Kong et al., 2019; Zuo et al., 2020; Cheng et al., 2022; Hu et al., 2023; Zheng et al., 2023) (Table 1).

Notably, changes in some taxa are inconsistent across human studies. This variability may arise from differences in patient populations (*e.g.*, etiology of ARDS, comorbidities), sample collection timing, sequencing methodologies, or statistical approaches (Li & Ma, 2021). In addition, opposite changes in gut microbiota have been observed in animal models of ARDS compared to human studies (*Eubacterium* being one example) (Cheng et al., 2022; Zheng et al., 2023). While animal models provide valuable mechanistic insights into gut-lung axis interactions in ARDS, the specific microbial shifts observed in these models may not fully recapitulate the dysbiosis patterns seen in human patients. Future studies with larger, more homogeneous cohorts are needed to clarify these associations.

Multiple studies demonstrate a significant reduction in α-diversity in the lung microbiome of ARDS patients compared to those without ARDS (Kyo et al., 2019; Schmitt et al., 2020; Imbert et al., 2024). However, findings exhibit considerable heterogeneity across studies involving ARDS of differing etiologies. Notably, Li et al. (2024) reported a progressive decline in microbial diversity within the pulmonary microbiome of septic ARDS patients over time, whereas Zhang et al. (2022) observed increased microbial diversity in patients with extrapulmonary infection-induced ARDS. Compared to the microecology

of a healthy lung, which is primarily dominated by *Streptococcaceae*, *Veillonellaceae*, *Prevotellaceae*, *Ruminococcaceae*, and *Flavobacteriaceae*, the lung microbiota of ARDS patients is predominantly composed of *Pasteurellaceae* and *Enterobacteriaceae* (*Panzer et al., 2018*; *Dickson et al., 2020*).

Animal studies indicate that the lung microbiome plays a key role in the pathogenesis of respiratory diseases. Its potential protective mechanisms may involve assisting the host in establishing and maintaining balanced immune homeostasis (*Martin-Loeches et al., 2020*). For instance, intranasal administration of *Acinetobacter lwoffii* to mice induces IL-6 and IL-10 production, thereby significantly alleviating allergen-induced airway inflammation (*Alashkar Alhamwe et al., 2023*). Conversely, studies in respiratory disease models have also revealed associations between the presence of *Ruminococcus gnavus* and the promotion of respiratory allergic responses (*Chua et al., 2018*). Notably, changes in the lung microbiota may influence the composition of the gut microbiota in patients with sepsis-induced ALI/ARDS. *Sze et al. (2014)* observed lung microbiota dysbiosis and an increase in total bacterial count in the cecum of mice in a sepsis-associated ALI mouse model. Studies have found that vancomycin treatment in mice with acute *Pseudomonas aeruginosa* pneumonia can induce gut microbiota dysbiosis, resulting in an increase in *Proteobacteria* and a decrease in *Bacteroidetes*, accompanied by changes in intestinal inflammation. These changes significantly improved after fecal microbiota transplantation (FMT) (*Rosa et al., 2020*). In addition to this, there have been some findings on the effect of lung inflammation on gut microbiota. The systemic inflammatory cascade in ARDS directly compromises the integrity of the intestinal barrier through TNF-α-mediated downregulation of TJ proteins (*e.g.*, claudin, occludin, ZO-1) (*Ziaka & Exadaktylos, 2024*). During sepsis-induced ARDS, increased proinflammatory factors can inhibit intestinal cell regeneration and promote apoptosis, leading to a reduction in intestinal mucosal layer thickness and barrier dysfunction (*Chen et al., 2018*). In a mouse model of influenza, influenza-induced interferon-I production in the lungs promotes the depletion of obligate anaerobes and the enrichment of *Proteobacteria* in the gut, creating a ''malnourished'' micro-environment (*Deriu et al., 2016*). Furthermore, ARDS-associated hypoxemia and hemodynamic instability can induce intestinal ischemia-reperfusion injury. This creates an anaerobic micro-environment conducive to pathogen overgrowth, thereby driving a shift toward intestinal dysbiosis (*Lv et al., 2024*).

Under normal conditions, the intestinal barrier consisting of intercellular junctions in the intestinal mucosal epithelia, permits the movement of water and immunoregulatory factors while preventing the passage of macromolecules and microorganisms (*Weström et al., 2020*). During sepsis, an increase in inflammatory cytokines leads to increased intestinal permeability by affecting the TJ between cells and the intracellular cytoskeleton. This results in the translocation of intestinal microorganisms, such as *Bacteroidetes* and *Enterobacteriaceae*, across the intestinal mucosa in patients with sepsis and ARDS, and even into the lungs (*Dickson et al., 2016*). Such bacterial translocation triggers localized activation of inflammatory mediators within the mucosal immune system. This process is intricately linked to the pathogenesis of ARDS, which involves dysregulated activation of inflammatory cytokine cascades (*Assimakopoulos et al., 2018*). The continued accumulation

of inflammatory cells attracts more proinflammatory factors, which interact to induce greater infiltration of inflammatory cells, thereby establishing a vicious cycle of lung injury (*Li et al., 2019*), in addition, lung inflammation may affect the structure of the gut bacterial community and further exacerbate lung inflammation (*Vital et al., 2015*). Notably, gut microbiota-derived metabolites also influence pulmonary immunity. Specifically, SCFAs modulate immune cell function and dampen excessive inflammation (*e.g.*, *via* NF-κB pathway inhibition) through mechanisms such as HDAC inhibition and activation of G protein-coupled receptors (GPR41, GPR43, GPR109a) (*Cox et al., 2009*). Furthermore, SCFAs may exert protective effects on alveolar epithelial barrier integrity (*Mukhopadhya & Louis, 2025*). However, both ARDS patients and animal models exhibit reduced intestinal and/or circulating SCFA levels. This decline compromises their protective roles in maintaining gut barrier function and suppressing pulmonary inflammation, thereby exacerbating disease severity (*Bezemer et al., 2024*; *Xuan et al., 2024*). Conversely, SCFA supplementation (*e.g.*, butyrate gavage) demonstrates protective effects in animal models, highlighting its therapeutic potential (*Diao et al., 2019*; *Mao et al., 2024*). Collectively, these mechanisms jointly establish a "gut-lung axis" of the injury cycle. Targeting this axis—particularly barrier integrity, dysbiosis, and bacterial translocation—represents a promising therapeutic frontier.

## Impact of intestinal microecology on ARDS

Studies have explored the roles of apoptosis, autophagy, pyroptosis, and oxidative stress in the pathogenesis of ARDS, but have typically examined these pathways in isolation or focused solely on lung inflammation (*Shi et al., 2022*). In this review, we systematically integrate these four mechanisms within the framework of the gut microbiota-ARDS axis and link these mechanisms to clinically actionable interventions in an attempt to address gaps in the purely mechanistic literature (Fig. 2).

Oxidative stress, an imbalance between pro-oxidants and anti-oxidants, plays a key role in the development of ARDS (*Sies, 2015*). *Tang et al. (2021)* have demonstrated that gut microbiota exerts a protective effect on regulating LPS-induced immune responses in ALI by modulating the TLR4/NF-κB signaling pathway, which may induce inflammation and oxidative stress. In an ARDS mouse model, LPS treatment significantly increases the levels of MDA, IL-6, and TNF-α, while decreasing the activity of SOD and GSH-PX43. Conversely, inhibiting ROS and enhancing the levels of SOD and glutathione in lung tissue help protect mice from ARDS infection (*Zhang et al., 2015*). These findings collectively suggest a complex interaction network between oxidative stress and gut microbiota.

Apoptosis is a critical factor in the development of ARDS. When changes occur in the gut microbiota during ARDS, these changes are often associated with increased apoptosis in lung and intestinal tissues (*Fleisher, 1997*). In a Staphylococcal enterotoxin B-induced mouse model of ARDS, researchers observed disruptions in both lung and gut microbiota, as well as an increased proportion of apoptotic cells (*Alghetaa et al., 2021*). Similarly, *Li et al. (2018)* observed a significant increase in the proportion of apoptotic cells in the ALI-BALB/c mouse model, accompanied by downregulation of Bcl-2 expression and upregulation of TNF-α, IL-6, IL-1β, Bax, and cleaved caspase-3. Apoptosis

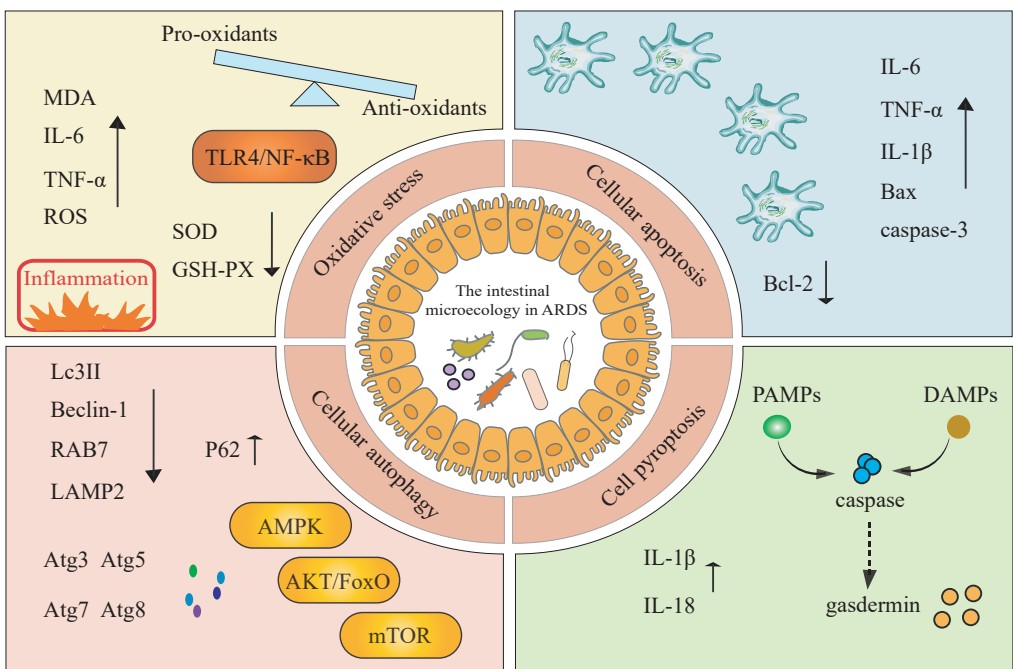

**Figure 2** **The modulation of gut microbiota impacts the onset and progression of ARDS by triggering oxidative stress, cellular apoptosis, cellular autophagy, and cell pyroptosis mechanisms.** MDA, malondialdehyde; SOD, superoxide dismutase; GSH-PX, glutathione peroxidase; ROS, reactive oxygen species; LPS, lipopolysaccharide; Atg, autophagy-related proteins; AKT, alpha serine/threonine kinase; FoxO, forkhead box O; AMPK, AMP-activated protein kinase; mTOR, mammalian target of rapamycin.

is triggered by two pathways: the intrinsic pathway involves regulators within the cell itself, including cytochrome c, cleaved caspase-3, Bax, and Bcl-2; and the extrinsic pathway associated with extracellular stimuli and apoptotic receptors on the cell membrane (*Shi et al., 2022*). Furthermore, Xu et al. (*Mustansir et al., 2021*) proposed that the administration of lysophosphatidylcholine can reduce monocyte infiltration in a mouse model of ARDS by modulating the MAPK/NF-κB signaling pathway and inhibiting monocyte apoptosis triggered by galactosamine-lipopolysaccharide, thereby improving survival rates and lung function.

Cell autophagy is a "double-edged sword," exerting protective effects under normal conditions but potentially becoming harmful when over-activated (*Peng et al., 2022*). Inducing autophagy can protect mice from ALI triggered by LPS and mechanical ventilation, and it can also improve arterial oxygenation and vascular function (*Nosaka et al., 2020*). In patients with sepsis combined with ARDS, the expression of autophagy-related proteins, such as LC3II, Beclin-1, RAB7, and LAMP2, is reduced in peripheral blood, while p62 is significantly increased. This suggests that autophagy is inhibited in these patients (*Xu et al., 2022*). Notably, the gut microbiota interacts with cellular autophagy to promote disease progression. *Cheng et al. (2018)* triggered intestinal mucosal autophagy through FMT and alleviated intestinal barrier damage in piglets caused by *Escherichia coli* K88. Various Atg, such as Atg8, Atg7, Atg5, and Atg3, as well as several signal transduction pathways, including

the AKT/FoxO, AMPK, and mTOR pathways, are involved in regulating autophagy at different stages of autophagosome formation. Another study in cellular models of ARDS and rat models of ALI confirmed that penehyclidine hydrochloride treatment enhances proliferation and autophagy in the ALI model, while reducing apoptosis and inflammation (*Wang et al., 2021*). However, there are currently few clinical studies on ARDS patients, and how the gut microbiota affects cellular autophagy in ARDS still lacks sufficient evidence.

In contrast to apoptosis, cellular pyroptosis is a novel type of programmed cell death associated with inflammation. Initially, PAMPs and DAMPs activate inflammasomes, which subsequently trigger the activation of caspases, ultimately leading to the cleavage of the gasdermin protein family (*Hsu et al., 2021*). Recent studies have shown that pyroptosis-related cytokines, such as IL-1β and IL-18, are significantly elevated in bronchoalveolar lavage fluid from ARDS patients, and previous research has demonstrated that specific inhibition of IL-1β and IL-18 expression can significantly reduce lung injury (*Peukert et al., 2021*). Furthermore, activated caspase-1 and gasdermin D have been detected in the serum of ARDS patients, which may trigger a cascade of pyroptosis (*Homsy et al., 2019*). Notably, some Gram-negative bacteria in the gut (*e.g.*, *Pseudomonas aeruginosa*) can initiate NLRC4 and caspase-11-dependent pyroptosis through flagellin and LPS (*Wei, Zhang & Song, 2022*). A decrease in intestinal *Parabacteroides merdae* during pregnancy can lead to weakened interaction with NLRP3, which promotes cellular pyroptosis and increases sepsis responses and tissue damage (*Chen et al., 2023*). Currently, there is limited research on the relationship between pyroptosis and the gut microbiota. Understanding the link between pyroptosis and alterations in the gut microbiota in ARDS is an emerging research area that may reveal new therapeutic approaches.

## Current management and emerging microbiome-targeted interventions for ARDS

Current cornerstones of ARDS management include lung-protective ventilation (notably low tidal volume), adjunctive therapies (such as prone positioning in moderate-to-severe cases), and conservative fluid management in the absence of shock, all supported by robust randomized controlled trial (RCT) evidence. However, targeted pharmacological interventions addressing immune dysregulation and inflammatory injury remain markedly lacking. Consequently, exploring novel therapeutic avenues, such as modulating the microbiome, represents a promising approach (*Meyer, Gattinoni & Calfee, 2021*).

Some studies have explored the application of certain bacterial species as biomarkers for the early diagnosis of respiratory diseases. *Chen et al. (2024)* revealed that circulating microbiome DNA can be a biomarker for the diagnosis and recurrence of lung cancer, with a high sensitivity of 87.7% and an AUC of 93.2% in an independent validation dataset for their diagnostic model. Similarly, *Marshall et al. (2022)* analyzed the respiratory microbiota profiles of nearly 400 patients and created and validated a microbiome-based classifier that can predict the likelihood of lung cancer in asymptomatic patients before clinical diagnosis. This study demonstrated the clinical potential of respiratory microbiota profiling for early lung cancer detection.

Given these limitations, microbiome-targeted interventions represent a promising frontier for addressing immune dysregulation in ARDS. It is noteworthy that probiotics have already been applied in the clinical treatment of diseases. Since the 1950s, FMT has made significant progress in correcting microbiota disorders, repairing intestinal barriers, and regulating immunity (*Wang et al., 2019*). Furthermore, several meta-analyses have also reported the success of FMT in the treatment and prevention of acute infectious diarrhea and upper respiratory infections (*Hempel et al., 2012*; *Popova et al., 2012*). A study in 2020 showed that intranasal irrigation with live *Lactobacillus lactis* W136 in patients with chronic rhinosinusitis significantly improved their sinus symptoms (including nasal obstruction, postnasal drip, and "needing to blow their noses"), quality of life, and mucosal scores (*Endam et al., 2020*). However, there are few reports on the direct use of gut microbiota in the diagnosis and treatment of ARDS. As early as 1972, the Cuevas team found prevention of ARDS by enteral antibiotic pretreatment in an animal model of shock (*Silvestri, De la Cal & Van Saene, 2012*). In a mouse model of ALI, FMT was used to reconstruct the gut microbiota, resulting in increased gut microbiota diversity and an increase in beneficial bacteria capable of producing SCFAs that counteract acute lung injury. This effectively inhibited the activation of the TLR4/NF-κB signaling pathway, inflammation, and the release of oxidative stress factors in the lungs of ALI animals (*Tang et al., 2021*). Another study reported a significant reduction in in-hospital mortality among high-risk patients who received prophylactic gastrointestinal decontamination therapy (*Hammond et al., 2022*). However, the application of intestinal microecology in the diagnosis and management of ARDS needs to be explored with more studies.

## Lessons from failed immunomodulatory trials and microbiome interactions

Despite extensive research into ARDS pathogenesis, numerous clinical trials targeting immune dysregulation have failed to improve outcomes, highlighting the disease's clinical and biological heterogeneity (*The ARDS Network Authors for the ARDS Network, 2000*; *Steinberg et al., 2006*; *Normand, 2020*). Notably, several agents designed to modulate host immunity showed promise preclinically but proved ineffective in RCTs.

For instance, an ARDS network trial conducted in 2006 demonstrated no mortality benefit with methylprednisolone, with potential harm in late-phase ARDS (*Steinberg et al., 2006*). It is known that steroids in non-ARDS diseases (*e.g.*, IBD) may cause or exacerbate dysbiosis, reduce microbial diversity, and increase the abundance of potentially pathogenic bacteria (*Selinger et al., 2017*). We speculate whether these known effects may be under-recognized in ARDS trials and may partially counteract their anti-inflammatory benefits or pose additional risks. In addition, the results of an RCT showed that ketoconazole as an anti-inflammatory agent did not reduce the incidence of ARDS in high-risk patients (*The ARDS Network Authors for the ARDS Network, 2000*). Although it failed as an anti-inflammatory agent in ARDS, it is a potent antifungal agent in its own right (*Akova et al., 2025*). Even if the purpose of the trial was anti-inflammatory, its significant perturbation of the intestinal fungal group (as well as the bacterial group affected by fungi) was unavoidable. We wonder whether this unintended, strong perturbation of the microbiome might be a

factor in its lack of efficacy or even potential harm. Otherwise, COVID-19 convalescent plasma has not shown any efficacy in patients with severe COVID-19 ARDS (*Normand, 2020*). Although plasma components (*e.g.*, antibodies) may alter the resilience of microbial communities, direct evidence in ARDS remains scarce (*Vogl et al., 2021*).

These failures underscore the complexity of ARDS immunopathology and the overlooked role of microbiome-immune crosstalk. While microbiome shifts during such interventions are poorly characterized in ARDS cohorts, studies in sepsis suggest that immunomodulatorscan exacerbate dysbiosis and bacterial translocation (*Lin et al., 2023*; *Wang et al., 2025*). Future trials should integrate microbiome monitoring to elucidate whether microbial dynamics contribute to treatment non-response and identify patient subsets benefiting from microbiota-targeted adjuvants.

## CONCLUSIONS

In conclusion, ARDS persists as a critical condition with high mortality despite optimized supportive care, necessitating novel therapeutic strategies. This review establishes intestinal microecology as a central regulator of ARDS pathogenesis through bidirectional gut-lung axis interactions, wherein ARDS-induced gut barrier dysfunction (involving TJ disruption, mucin depletion, and antimicrobial peptide suppression) drives dysbiosis and bacterial translocation, while gut-derived metabolites (notably SCFAs) and dysbiotic microbiota exacerbate lung injury *via* oxidative stress, apoptosis, autophagy, and pyroptosis. Although microbiome-targeted interventions (*e.g.*, FMT, probiotics) show preclinical promise, lessons from failed immunomodulatory trials (*e.g.*, corticosteroids, ketoconazole) reveal that microbiome-immune crosstalk is a critical yet understudied determinant of treatment efficacy. Moving forward, resolving inconsistencies between animal models and human data through standardized longitudinal sampling, multi-omics integration, and validation of gut microbiota signatures as early biomarkers will be essential to advance microbiota-directed therapies. By bridging these mechanistic insights with clinical translation, targeting the gut-lung axis represents a paradigm shift toward precision management of ARDS.

**Abbreviations**

| | |
|---|---|
| **ALI** | acute lung injury |
| **AKK** | Akkermansia muciniphila |
| **AKT** | alpha serine/threonine kinase |
| **AMPK** | AMP-activated protein kinase |
| **AMPs** | antimicrobial peptides |
| **ARDS** | acute respiratory distress syndrome |
| **Atg** | autophagy-related proteins |
| **DAMPs** | damage-associated molecular pattern |
| **HDAC** | histone deacetylase |
| **FMT** | fecal microbiota transplantation |
| **FoxO** | forkhead box O |
| **GSH-PX** | glutathione peroxidase |
| **HMGB1** | high-mobility group box 1 protein |

| ICU | intensive care unit |
| IFN-γ | interferon-γ |
| IL | interleukin |
| LPS | lipopolysaccharide |
| MAPK | mitogen-activated protein kinase |
| MDA | malondialdehyde |
| mTOR | mammalian target of rapamycin |
| NF-κB | nuclear factor-κB |
| PAMPs | pathogen-associated molecular patterns |
| RCT | randomized controlled trial |
| ROS | reactive oxygen species |
| SCFAs | short-chain fatty acids |
| SOD | superoxide dismutase |
| TJ | tight junction |
| TLR | toll-like receptor |
| TNF-α | tumor necrosis factor-α |
| ZO-1 | zonula occludens-1 |

### Funding
The authors received no funding for this work.

### Competing Interests
The authors declare there are no competing interests.

### Author Contributions

- Lujia Chen conceived and designed the experiments, performed the experiments, analyzed the data, prepared figures and/or tables, authored or reviewed drafts of the article, and approved the final draft.
- Hao He analyzed the data, prepared figures and/or tables, authored or reviewed drafts of the article, and approved the final draft.
- Cheng Li analyzed the data, authored or reviewed drafts of the article, and approved the final draft.
- Sha Nie analyzed the data, prepared figures and/or tables, authored or reviewed drafts of the article, and approved the final draft.
- Dong Liu analyzed the data, prepared figures and/or tables, authored or reviewed drafts of the article, and approved the final draft.
- Qiwei Shi conceived and designed the experiments, performed the experiments, analyzed the data, authored or reviewed drafts of the article, and approved the final draft.

### Data Availability
This is a literature review.

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
