# Peer review of "A mini-review of the relationship between intestinal microecology and acute respiratory distress syndrome"

_PeerJ, doi:10.7717/peerj.19995_

## Round 0.1 · original submission · Major Revisions

Please address all the concerns raised by the reviewers.

·

Basic reporting

Keywords of this review need to be mentioned.
Regarding the figures the references they adopted from to be mentioned under them.
There are two References sections to correct this.

Experimental design

No comment

Validity of the findings

No comment

Reviewer 2 ·

Basic reporting

-

Experimental design

Fulfills most study design requirements. Need further granularity on search strategy - see noted below.

Validity of the findings

-

Additional comments

ARDS is a clinical syndrome with high heterogeneity, and dysbiosis in the gut microbiome has been previously characterised. In conjunction with this, experimental manipulation of gut microbial community (specifically FMT) has been proposed for its theoretical benefit for patients with acute lung injury. In this review by Chen et al., the authors reviewed key inflammatory signalling pathways and described the change in microbial landscape in the gut microbiome.

Major point –
1. Search strategy – please kindly provide more details (consider including a flowchart for inclusion and exclusion of papers). Would also suggest including gut microbiome** besides microbiota.

2. Lack of clarity on dysbiosis in the setting of ARDS – consider adding a diagram summarising which taxa are enriched in ARDS and vice versa per each paper. There should be many characterisation papers out there, and they do not necessarily agree with each other.

3. Microbiome in gut versus other body compartments – maybe I was reading it wrong, and I know microbiome outside the gut is not necessarily the focus of this study, but I am afraid that airway microbiome research is not well represented in this paper, given its importance. My impression is that there are many papers on multicompartment characterisation of clinical specimens from ARDS patients, as well as some basic studies on shift in SCFA level and dysbiosis in both airway and gut microbiome.

4. Consider dedicating another paragraph to a comprehensive review of failed trials targeting ARDS immune dysregulation and their impact on the microbiome. Given its clinical and biological heterogeneity, most of the proposed medications for ARDS have failed RCTs, among which some are agents that target host immune, including steroids (NEJM 2006), ketoconazole (not as an anti-fungal agent but as an anti-inflammatory agent, JAMA 2000), convalescent plasma for COVID ARDS (NEJM 2020) and many more. Please consider updating your search strategy and try to include more papers describing the shift in microbiome in a population undergoing those interventions, if there is any.

5. On top of the point above, I noticed that the author claimed that ‘Currently the treatment of ARDS mainly relies on supportive monitoring’ (line 252), which I personally found unfair for the solid trials on low tidal volume ventilation, prone positioning, restrictive fluid strategy, etc. These interventions are not merely supportive monitoring. I would kindly suggest the author revise such a statement throughout.

6. Need further synthesis of materials. It is clear to me that the authors have spent a lot of effort and are well-versed in contemporary basic research on ARDS, and this manuscript is well written with only a few typos or grammatical errors. However, more often than not, I found the authors are merely listing summary sentences for a tapestry of marginally relevant studies in one paragraph and lack organic synthesis. If possible, please consider rearranging the section ‘Impact of ARDS on intestinal microecology’.

Minor point -
1. Abstract – ‘This article provides valuable insights for clinicians, particularly those in intensive care units’. Consider tuning down the language, or please kindly explain in detail how exactly this will directly impact ARDS clinical management in the ICU.
2. Cover page, address section - what does ‘Vic’ in ‘Hengyang, Vic 421000’ mean? Is that part of the zip code?

·

Basic reporting

The review manuscript titled (A mini-review of the relationship between intestinal
microecology and acute respiratory distress syndrome) focuses on Knowledge for clinical researchers, those researching severe respiratory conditions, microbiology, and doctors, especially those working in intensive care units. The microbiome is a crucial subject in all fields, particularly in the context of respiratory diseases.
Major issues
1- There are more subjects to cover within this review, like intestinal permeability, which includes (tight junctions) Occluding, Zonula-1, and claudin. This should all be covered in this review.
2- The review should also focus on antimicrobial peptides and proteins.
3- The review should also focus on the mucin produced by the bacteria.
4- Short-chain fatty acids are a very important subject and related to microbes and ARDS.
5- All the subjects above have been studied in many research studies, and one of them you mention in line 206, has extensive work on all of those subjects.

Experimental design

-

Validity of the findings

-

---

## Round 0.2 · Major Revisions

Please address all reviewers comments.

·

Basic reporting

The review article with the title (A mini review of the relationship between intestinal
microecology and acute respiratory distress syndrome). Focused on intestinal microecology as a key modulator of acute respiratory distress syndrome development via mutually beneficial relationships with the gut-lung axis,
Major issues
1- Lines 130-136 have unrelated information to acute respiratory distress syndrome and are old information. Remove these lines.
2- Line 334: (The microecology-based interventions for ARDS) is not a suitable title because from lines 335-350, about complications, and lines 351- 370 are about FMT. Change the title.
3- FMT in line 352 is only an abbreviation. You should write the full name when it comes for the first time.
4- Add a list of abbreviations.

Experimental design

no comment

Validity of the findings

no comment

---

## Round 0.3 · accepted · Accept

Thanks for addressing all of the reviewers' comments.

·

Basic reporting

No comment

Experimental design

No comment

Validity of the findings

No comment